# Identification and Functional Analysis of AopN, an *Acidovorax Citrulli* Effector that Induces Programmed Cell Death in Plants

**DOI:** 10.3390/ijms21176050

**Published:** 2020-08-22

**Authors:** Xiaoxiao Zhang, Mei Zhao, Jie Jiang, Linlin Yang, Yuwen Yang, Shanshan Yang, Ron Walcott, Dewen Qiu, Tingchang Zhao

**Affiliations:** 1State Key Laboratory for Biology of Plant Diseases and Insect Pests, Institute of Plant Protection, Chinese Academy of Agricultural Sciences, Beijing 100193, China; zhangxiao0719@126.com (X.Z.); jiangjie0427@126.com (J.J.); yanglinlin0330@163.com (L.Y.); yangyuwen@126.com (Y.Y.); qiudewen@caas.cn (D.Q.); 2Department of Plant Pathology, University of Georgia, Athens, GA 30602, USA; mzhao@uga.edu (M.Z.); rwalcott@uga.edu (R.W.); 3Institute of Medicinal Plant Development, Chinese Academy of Medical Sciences and Peking Union Medical College, Beijing 100193, China; yangshanshan12@126.com

**Keywords:** AopN, programmed cell death, *Acidovorax citrulli*, plant pathogen, effector-triggered immunity

## Abstract

Bacterial fruit blotch (BFB), caused by *Acidovorax citrulli*, seriously affects watermelon and other cucurbit crops, resulting in significant economic losses. However, the pathogenicity mechanism of *A. citrulli* is not well understood. Plant pathogenic bacteria often suppress the plant immune response by secreting effector proteins. Thus, identifying *A. citrulli* effector proteins and determining their functions may improve our understanding of the underlying pathogenetic mechanisms. In this study, a novel effector, AopN, which is localized on the cell membrane of *Nicotiana benthamiana,* was identified. The functional analysis revealed that AopN significantly inhibited the flg22-induced reactive oxygen species burst. AopN induced a programmed cell death (PCD) response. Unlike its homologous protein, the ability of AopN to induce PCD was dependent on two motifs of unknown functions (including DUP4129 and Cpta_toxin), but was not dependent on LXXLL domain. More importantly, the virulence of the *aopN* mutant of *A. citrulli* in *N. benthamiana* significantly decreased, indicating that it was a core effector. Further analysis revealed that AopN interacted with watermelon ClHIPP and ClLTP, which responds to *A. citrulli* strain Aac5 infection at the transcription level. Collectively, these findings indicate that AopN suppresses plant immunity and activates the effector-triggered immunity pathway.

## 1. Introduction

Plant immune responses to pathogen infections involve two layers of immunity: pathogen-associated molecular pattern (PAMP)-triggered immunity (PTI) and effector-triggered immunity (ETI) [1]. When PAMPs are recognized by the PRR (pattern recognition receptor) on the plant cell surface, the PTI response is initiated [2]. The PTI response is often accompanied by active oxidative burst, callose deposition, and pathogenesis-related (PR) gene expression [3]. Plant pathogens often secrete effector proteins that interfere with plant PTI responses [4], leading to rapid pathogen propagation [5]. The interaction between plants and pathogens is synergistic [6]. Plants have also evolved resistance genes (R genes) that recognize effectors to trigger ETI. The resistance responses during this process are intense and often accompanied by a hypersensitive reaction and a rapid process of programmed cell death (PCD) that limits the further pathogen propagation [3]. However, pathogens, especially some plant pathogenic bacteria, also secrete certain effectors to suppress ETI [3]. For example, *Pseudomonas syringae* type III effector HopD1 inhibits ETI signaling but not PTI signaling [7]. Thus, the interaction between pathogen effectors and the plant immune system is complex and multilayered [8]. Previous studies have revealed the functions of some effector proteins in plant pathogenic bacteria and their interference with plant immunity [9,10]. For example, some effector proteins from *Pseudomonas syringae* pv. *tomato* DC3000 (*Pst* DC3000) inhibit the PTI response by inhibiting reactive oxygen species (ROS) burst, for example HopG1 [11]. Some effectors are reported to modulate environment conditions. For example, *Pst* DC3000 secretes HopM1, which contributes to virulence by establishing an aqueous living space in plants [12]. In addition, some effectors can induce PCD responses in the model plant *N. benthamiana* [13]. For example, effectors from *Xanthomonas oryzae* pv. *oryzicola*, namely, XopN, XopX, XopA, XopY, XopF1, and AvrBs2 [14], and effectors AvrE1, HopM1, HopQ1-1, HopR1, HopAM1, and HopAD1 from *Pst* DC3000 [11], were reported to induce PCD responses in *N. benthamiana* leaves. These PCD-inducing effectors often play an important role in interacting with the plant immune system. For example, the PCD-inducing XopX from *X. euvesicatoria* strain 85-10 contributed to virulence in plant as a key effector [8]. The PCD-inducing effector AvrBs2 along with 22 putative effectors from *X. oryzae* pv. *oryzicola* is an essential virulence factor for rice infection [14]. *Pst* DC3000 (lacking PCD-inducing gene *hopQ1-1*) became pathogenic with *N. benthamiana* [15]. Recently, three putative effectors of *A. citrulli* were found to induce PCD in *N. benthamiana* leaves. Among them, *A. citrulli* strains M6 and AAC00-1 mutated in the *Aave_1548* effector gene and displayed reduced virulence in *N. benthamiana* [16], which was the first detailed report of a putative PCD-inducing effector in *A. citrulli*. In a previous study, we found that the YopJ homologous effectors in *A. citrulli* Aac5 strain also induced PCD in *N. benthamiana* [17]. The findings of these studies indicate that these PCD-inducing effectors are extremely important for the virulence of plant pathogenic bacteria. 

Bacterial fruit blotch (BFB), caused by the Gram-negative bacterium *A. citrulli* [18], is prevalent worldwide [19], resulting in significant economic losses of cucurbit crops. The pathogenic mechanism of *A. citrulli* is unclear, and effective prophylactic and control measures of BFB are lacking. Only a few studies have examined the effector proteins in *A. citrulli*, and the plant immune responses triggered by *A. citrulli*. Host resistance-based measures are the most effective disease control strategy in the field [20]. Thus, identifying and analyzing functions of effector proteins of *A. citrulli* are essential in providing a foundation for studies on host resistance against BFB. In recent years, some progress has been made in *A. citrulli* effector proteins [21]. For example, *A. citrulli* group I and II strains differ in their effector compositions [22], and the presence or absence of these group-specific effectors may determine the preference of *A. citrulli* strains for different cucurbit hosts [23]. As one of the natural hosts of *A. citrulli* is watermelon, and the immune pathway of watermelon is not well-studied, it is difficult to analyze the functions of *A. citrulli* effector proteins in watermelon. Recent studies have shown that the interaction system of *A. citrulli* and *N. benthamiana* can be used for the functional analysis of *A. citrulli* effector proteins [16]. HrpG and HrpX were previously identified as key regulatory factors of the type III secretion system (T3SS) in *A. citrulli* [17]. Using this *hrpX* mutant, we screened and identified an effector protein, AopP, which interfered with plant immune responses (unpublished). A recent study also used *hrpX* from *A. citrulli* group I strain M6 to screen for effector proteins using the transcriptome analysis [21]. However, only a few studies have been conducted to identify and characterize *A. citrulli* effector proteins.

In this study, the effector protein AopN in *A. citrulli* strain Aac5 was identified and found to be localized in the cell membrane. It induced PCD in *N. benthamiana*. More importantly, the virulence of *N. benthamiana* significantly reduced when the *aopN*-coding gene was knocked out. This study provides new insights into the virulence mechanism of *A. citrulli.*

## 2. Results

### 2.1. Sequence Analysis of Putative T3E AopN from A. Citrulli

The XopN homologous effector protein in *A. citrulli* was discovered by BLAST analysis. The putative effector (AopN) in the Aac5 strain contains 635 amino acids, with a predicted molecular size of 68.01 kDa, and its coding gene is 100% homologous to *Aave_3621* of the AAC00-1 strain. Importantly, the Kyoto Encyclopedia of Genes and Genomes (KEGG) database analysis revealed two motifs in AopN, namely, DUF4129 and Cpta_toxin (Figure 1a). The DUF4129 motif contains 47 amino acids from 20 to 66 and the Cpta_toxin motif contains 43 amino acids from 215 to 257. The BLAST analysis showed the most similar homolog (93.86%) of AopN from *A. citrulli* was AopN from *Acidovorax cattleyae* (WP_092832167). Amino acid sequence alignment of these proteins is presented in Figure 1b.

### 2.2. AopN is A Type 3 Effector in A. Citrulli

To examine whether AopN is in the regulon of HrpX in *A. citrulli*, quantitative PCR (qPCR) was performed. The wild-type (WT) strain Aac5 and the *hrpX* mutant were induced in the T3SS-inducing medium; the expression of *aopN* in the *hrpX* mutant was significantly lower than that in the WT, indicating that *hrpX* regulated *aopN* at the transcription level (Figure 2a). Based on this, the secretory function of AopN was analyzed by Western blotting. AopN-fused FLAG coding sequences driven by its native promoter was constructed and transformed into the *A. citrulli* WT Aac5 strain and its T3SS-deficient strain *hrcJ* mutant. These strains were induced in the T3SS-inducing medium, after which the intracellular and exocrine components were extracted. The Western blotting results revealed that the AopN-FLAG-fused protein was detected intracellularly in the WT and *hrcJ* mutant, and also in the supernatant of WT expressing the AopN-fused FLAG, whereas no signal was detected in the culture supernatant of the *hrcJ* mutant expressing AopN-FLAG (Figure 2b). The results indicated that AopN is a type 3 effector dependent on T3SS in *A. citrulli.*

### 2.3. AopN Contributes to Virulence in N. Benthamiana But Not in Watermelon

To examine the role of *aopN* in virulence, an *aopN* mutant strain was constructed and syringe-infiltrated into *N. benthamiana* and watermelon. The *aopN* mutant did not significantly reduce the virulence in watermelon cotyledons compared to the WT and its complemented strain (Figure 3b). Further quantitative analysis of bacterial population levels also showed that there was no significant difference in the growth levels between WT Aac5 strain, *aopN* mutant, and its complemented strain 2 and 4 days after inoculation (Figure 3a). In addition, the role of *aopN* in virulence on *N. benthamiana* was analyzed. Interestingly, the *aopN* mutant showed significantly reduced virulence in the infected leaves of *N. benthamiana* compared with the WT strain, which exhibited symptoms similar to the leaves infected with the *hrcJ* mutant (Figure 3c). These findings indicate that AopN contributes to virulence in *N. benthamiana* but not in watermelon.

### 2.4. AopN was Localized at the Plant Cell Membrane.

AopN-GFP (green fluorescent protein) fluorescence was observed at the cell membrane. The GFP control was observed in the entire cells and the fluorescence signal generated by the plasma membrane marker carrying the red fluorescent protein was superimposed on the fluorescence signal generated by AopN-GFP (Figure 4), confirming that it is indeed localized at the membrane.

### 2.5. AopN Induced PCD in N. Benthamiana

To examine whether AopN induced PCD in *N. benthamiana*, the leaves of *N. benthamiana* were injected with 35S: AopN-FLAG or empty vector at an optical density at 600 nm (OD_600_) of 0.5. After 3 days, PCD was evaluated in the collected leaves. The areas where the leaves were injected with GV3101 expressing AopN exhibited necrosis, whereas GV3101 carrying the empty vector did not show necrotic areas (Figure 5a). For further analysis, the conductivity test was performed. In the areas where the leaves were injected with GV3101, the expression of AopN was significantly increased compared with that of the control (Figure 5b). To further verify that the necrotic symptoms were caused by AopN expression, Western blotting assays were performed. The results confirmed that AopN was expressed in those areas of the leaves collected after transient expression after 36 h and 48 h at OD600 = 0.3 (Figure 5c). These results indicated that AopN induced PCD in plants and triggered the ETI.

### 2.6. AopN Suppressed Reactive Oxygen Species (ROS) Burst in N. Benthamiana

To examine whether AopN affected ROS burst, an flg22-induced ROS assay was performed. Since the early PCD response of plant can also trigger ROS [24], it is necessary to find a suitable expression time point to detect this effect. The electrolyte leakage is an important indicator to measure the PCD response of plants. Based on the above results (Figure 5b), AopN induced the electrolyte leakage of *N. benthamiana* at 24 h and showed no significant difference compared to empty vector (EV) control using *A. tumefaciens* GV3101 at an optical density at 600 nm (OD_600_) of 0.5. At this point in time, the AopN fused FLAG can be expressed in *N. benthamiana* by Western blot (Figure 5c). Therefore, it is reasonable to test whether AopN affects ROS under this condition (24 h, *A. tumefaciens* GV3101 at an optical density at OD_600_ of 0.5). The signal intensity generated by 35S: AopN-FLAG treatment during 40 min was significantly lower than that of the control treatment (empty vector) (Figure 6a), when expressed in *N. benthaamiana*. To further analyze the effect of ROS inhibition, the PTI marker gene expression levels were detected by qPCR with flg22 treatment or no-flg22 treatment. The results showed that the expression levels of *NbPti5*, *NbAcre31*, and *NbGras2* were significantly reduced when 35S: AopN-FLAG was expressed compared with that under the empty vector control treatment with flg22 treatment (Figure 6b). Importantly, the expression levels of *NbPti5*, *NbAcre31*, and *NbGras2* were also significantly reduced when 35S: AopN-FLAG was expressed compared with that under the empty vector control treatment without flg22 treatment (Appendix A). These results indicate that AopN suppressed the PTI pathway by regulating the ROS burst in *N. benthamiana*.

### 2.7. Analysis of Key Motifs of AopN Inducing PCD

As the results showed that AopN induced PCD in *N. benthamiana*, the motifs responsible for triggering the PCD response were analyzed further. A previous study showed that LXXLL in the AopN homologous protein XopN is a key motif involved in its toxic function, and the bioinformatic analysis in the current study showed that AopN also had an LXXLL motif (Figure 7a). Thus, a construct with a point mutation in the LXXLL motif in AopN was generated (named AopN-LXXLL), and the construct was transformed into the GV3101 strain. The strain expressing AopN-LXXLL induced PCD in *N. benthamiana*, indicating that the LXXLL motif in AopN did not affect PCD (Figure 7b). AopN contained two motifs, DUF4129 and Cpta_toxin (Figure 1a). Constructs were made that lacked the DUF4129 and Cpta_toxin motifs, and these constructs were transformed into the GV3101 strain. The results showed that 35S:AopN-FLAG without DUF4129 and Cpta_toxin motifs expressed in *N. benthamiana* lost their ability to induce the PCD of *N. benthamiana* compared to that wild-type protein 35S:AopN-FLAG, indicating that the DUF4129 and Cpta_toxin motifs play important roles in regulating PCD in *N. benthamiana* (Figure 7c).

### 2.8. AopN Interacted with ClHIPP and ClLTP

The results revealed that AopN inhibited *N. benthamiana* PTI and triggered ETI. The interaction targets in the natural host watermelon were examined. To identify the potential interacting proteins with AopN, a yeast two-hybrid screen library, which was constructed with watermelon by *A. citrulli* strain Aac5 induction, was prepared using the *aopN* fragment as the bait. A putative heavy metal transport protein encoded by Cla97C06G115330 from watermelon genome 97103, named as *ClHIPP* was identified based on yeast two-hybrid screening (Appendix A). In addition, a putative Lipid transfer protein encoded by Cla97C10G187590 from watermelon genome 97103, named as *ClLTP*, was also identified based on yeast two-hybrid screening (Appendix A). To verify the interaction, the full-length sequences of *ClHIPP* and *ClLTP* were cloned and fused with the cYFP tag to perform the bimolecular fluorescence complementation (BiFC) assay. Figure 8 demonstrates a yellow signal, and the plasma membrane fuel with a red signal can overlap it. No fluorescence signal was observed with the negative control. The results indicated that AopN interacted with ClHIPP and ClLTP in vivo, and the interaction site was at the cell membrane. 

### 2.9. The Expression of ClHIPP and ClLTP Responds to A. Citrulli Strain Aac5 Infection

To further clarify whether the two interacting targets responds to *A. citrulli* infection. An flg22-induced assay was performed. The flg22*^Ac^* from FliC of *A. citrulli* was use. The watermelon leaves were sprayed with flg22*^Ac^* and the qPCR assay was performed. The expression levels of *ClLTP* were not associated with flg22*^Ac^*, but the expression levels of *ClHIPP* were associated with flg22*^Ac^* (Figure 9b). Further analysis shows that the expression levels of *ClLTP* and *ClHIPP* were associated with 0.1 mM salicylic acid (SA) treatment (Figure 9c,d). In order to more accurately assess the impact of this transcription level, watermelon cotyledons are sprayed with *A. citrulli* strain Aac5 at 3 × 10^8^ CFU/mL. In this study, total RNA of watermelon cotyledons at eight different time points during Aac5 infecting were extracted, and then two genes containing *ClLTP* and *ClHIPP* were tested by qPCR technology. The expression levels of *ClLTP* at 0.5 h, 1 h, 2 h, 4 h, 6 h, 24 h, 48 h after Aac5 infected significant differences compared to the expression levels of *ClLTP* at 0 h (Figure 9e), which is up-regulated at 2 h, 4 h, 48 h time points, and down-regulated at 0.5 h, 1 h, 6 h, 24 h time points. Similarly, the expression levels of *ClHIPP* at 0.5 h, 1 h, 2 h, 4 h, 6 h, 24 h, 48 h after Aac5 infected significant differences compared to the expression levels of *ClHIPP* at 0 h (Figure 9f), which is up-regulated at 0.5 h, 1 h, 6 h, 24 h time points, and down-regulated at 2 h, 4 h, 48 h time points. As flg22 triggers plant immunity [25], and salicylic acid (SA) is closely related to plant immune response [26], the flg22*^Ac^* treatment and SA treatment support this conclusion (Figure 9e,f). These findings indicate that *ClHIPP* and *ClLTP* responds to *A. citrulli* strain Aac5 infection. 

## 3. Discussion

Only a few studies have examined type III secreted effectors of *A. citrulli*. In this study, a novel effector, AopN, was identified. The virulence function, localization, and immune-related function of AopN was investigated to further understand the virulence of *A. citrulli*. T3Es have been reported to induce PCD in *N. benthamiana* [8,14]. Among these PCD-inducing T3Es, the XopN homologs from phytopathogenic *Xanthomonas* spp. are generally conserved, as determined by genome analysis [27]. Most of the predicted T3Es of *A. citrulli* shared similarities with those of *Xanthomonas* spp., as determined by bioinformatic analyses [21]. In this study, the homologous protein XopN in the *A. citrulli* Aac5 strain was identified by BLAST analysis and named AopN; the result is similar to that reported in the *A. citrulli* AAC00-1 strain and the M6 strain [21]. Although AopN and XopN are homologous, they have just 26% similarity [27], and AopN was classified into subgroup III compared with *Xanthomonas* spp. and *P. syringae* pv. *phaseolicola* 1448A [27]. AopN from *A. citrulli* had two motifs, including DUF4129 and Cpta_toxin, based on the KEGG database analysis, indicating that the motifs may have other functions related to virulence. Furthermore, AopN has homologous proteins in other *Acidovorax* spp., indicating that AopN may also be an important functional protein selected naturally during the evolution of *A. citrulli.*

Through bioinformatic analysis, the AopN homologs were predicted to be a T3E in *A. citrulli* strain M6 [21]. It has been determined, by transcriptome data analysis that the expression of the AopN-coding gene was downregulated in *hrpX* mutant of M6 compared with that in the wild-type strain M6 [21]. In this study, it was confirmed that the AopN-coding gene was downregulated in the *hrpX* mutant compared with the WT Aac5 strain, consistent with the findings of a previous study [21]. Our previous study indicated that *hrpX* was a core regulatory gene of T3SS in *A. citrulli* Aac5 strain [17], and Jimenez-Guerrero et al. (2020) [21] also confirmed that some candidate T3Es can be found, based on the difference in expression between the *hrpX* mutant and WT. Here, it was experimentally confirmed that AopN secretion was induced in the T3SS-inducing medium and was dependent on a functional type 3 secretion system (Figure 2b). These functional characteristics have been reported for numerous plant pathogenic bacteria based on Western blotting results, for example AvrXCCB [28]. 

It is well known that the functions of many T3Es in phytopathogenic bacteria are redundant. For instance, the T3Es AvrPto and AvrPtoB from *Pseudomonas syringae* pv. *tomato* DC3000 (*Pst* DC3000) strain have redundant functions [29]. Therefore, when a WT strain of phytopathogenic bacteria lacks a single effector, the virulence may not be substantially altered. For example, the deletion of only the *avrBs2* affected the pathogenicity of *X. oryzae* pv. *oryzicola* strain RS105, whereas deleting other effector-coding genes did not affect its pathogenicity in rice [14]. In this study, AopN was found to be a key contributor to virulence in *N. benthamiana* (Figure 3c). In a previous study, the *xopN* mutant did not affect the virulence of *X. oryzae* pv. *oryzicola* strain RS105 in rice [14], but *xopN* was a key contributor to the virulence of *X. oryzae* pv. *oryzicola* strain GX01 [30]. In the current study, the *aopN* mutant did not affect the pathogenicity of *A. citrulli* strain Aac5 in its natural host watermelon (Figure 3a,b), suggesting that *A. citrulli* may have other functional redundant proteins during watermelon infection. In a previous study, the XopN homolog was found to be functionally redundant with XopZ and XopV [31]. These findings also indicate that the interaction between AopN and *N. benthamiana* is important to elucidate the pathogenic mechanisms of *A. citrulli*.

T3E protein localization is important for the function of T3Es [32]. As previously reported, XopN is localized at the plasma membrane (PM) [33]. In the current study, AopN was also observed to be localized at the PM when expressed in *N. benthamiana.* This indicates that AopN and XopN have similar functions from the perspective of protein localization. Interestingly, AopN induced PCD symptoms, similar to XopN which was also reported to induce PCD symptoms [14]. Kim et al. [33] reported that the LXXLL motif of XopN played a key role in virulence. The bioinformatic analysis showed that the LXXLL motif was also present in AopN of *A. citrulli*. However, the LXXLL motif in AopN was not a key functional site and the AopN-LXXLL mutant still induced PCD symptoms, indicating that the molecular mechanism of AopN-induced PCD differed from that of XopN-induced PCD in *N. benthamiana.* Interestingly, AopN-induced PCD was related to the DUF4129 and Cpta_toxin motifs. These two motifs have not been widely examined, and it is unclear how the PCD-induction effects are exerted. 

The signaling of PTI and ETI is very complicated [34]. In a previous study, some genes involved in plant immunity are activated by PTI alone, and PTI plus ETI at early stage of RRS1/RPS4-mediated immune activation [35]. Pathogens can usually stimulate PTI and ETI of plants, especially some effectors [11]. Recent reports also show that in the absence of pathogens, the ETI related to leucine-rich repeat receptor (NLR), such as activation of RRS1 and RPS4, cannot activate mitogen-activated protein kinases (MAPKs) but still enhance expression of defense genes [36]. These continuously improving cognitions have enriched our cognitive ability of PTI and ETI. In particular, in plants, PTI is the first line of defense. When PTI is suppressed, resistance in plants is induced through ETI [3]. Some T3Es induce PCD to suppress the PTI pathway [11]. In a previous study, it was found that the YopJ homologous protein from *A. citrulli* Aac5 induced PCD symptoms [17]. Traore et al. [16] also reported that the YopJ homologous protein from *A. citrulli* AAC00-1 induced PCD symptoms in *N. benthamiana*. ROS burst is a key event in the PTI response [11]. In a previous study, the early stage of PCD in plants triggered ROS [24], for example, ROS was triggered in rice during the salt stress-induced early stage of PCD [37]. In the current study, the electrolyte leakage of *N. benthamiana* was performed to determine the appropriate detection time point, because the electrolyte leakage is an important indicator to measure the PCD response of plants [38]. Similarly, the electrolyte leakage assay was used for detecting the PCD induced by XopX in *N. benthamiana* [8]. In the current study, AopN from *A. citrulli* Aac5 induced PCD symptoms in *N. benthamiana,* and also suppressed the flg22-induced ROS burst in *N. benthamiana* (Figure 6a). More importantly, the expression levels of PTI marker genes containing *NbPti5*, *NbAcre31*, and *NbGras2* were also significantly downregulated by AopN whether it is flg22 treatment or not (Figure 6b, Appendix A). Overall, AopN inhibited the PTI response but stimulated the ETI response.

The identification of interaction targets provided insights into the molecular mechanism of *A. citrulli* pathogenicity. As AopN suppressed plant immunity, the molecular targets of AopN in the natural host watermelon were identified. The results showed that *ClHIPP* and *ClLTP* from watermelon interacted with AopN from *A. citrulli* (Figure 8, Appendix A). In past studies, PAMP flg22 triggered a plant immune response [25], and salicylic acid mediated resistance mechanisms are often associated with bacterial infection [39]. Recently salicylic acid in watermelon has also been reported to be involved in resistance to nematode infection [40]. To the best of our knowledge, flg22*^Ac^* from FliC of *A. citrulli* was used. The flg22*^Ac^*-induction treatment also showed that the *ClHIPP* was associated with PAMP flg22*^Ac^*. In addition, *ClLTP* and *ClHIPP* were associated with SA treatment. Importantly, these results support the expression of *ClLTP* and *ClHIPP* responds to *A. citrulli* infection.

## 4. Materials and Methods 

### 4.1. Plant Material and Bacterial Strains

Watermelon plants (cultivar Ruixin) are typically grown at 25 °C during the light cycle (16 h) and 20 °C during the dark cycle (8 h) under 40–60% relative humidity (RH) for bacterial inoculation assays. Tobacco plants are grown at 25 °C during the light cycle (16 h) and 22 °C during the dark cycle (8 h) under 40–60% RH for transient expression assays. *Escherichia coli* strain DH5α was cultured at 37 °C in Luria broth (LB). *Agrobacterium tumefaciens* GV3101 strain was grown at 28 °C in LB for transient expression. The *A. citrulli* WT strain Aac5 and its derivative strains were cultured at 28 °C in King’s B (KB) and T3SS-inducing broth [17]. All strains used in the assay are listed in Appendix A.

### 4.2. Construction of the AopN Marker-Less Mutant in A. Citrulli Aac5 

To generate the *aopN* marker-less mutant, a 503 bp fragment upstream of the *aopN* open reading frame (ORF) and a 455 bp fragment downstream of the *aopN* ORF were amplified by PCR and introduced into the pK18mobsacB vector [41]. The construct was transformed into the *A. citrulli* WT Aac5 strain. The markerless mutant was screened as previously described [17]. An *aopN-*complemented strain was constructed as described previously [17]. All primers used in the PCR are listed in Appendix A.

### 4.3. Construction of the AopN Mutant Protein Vector 

To generate the AopN mutant protein, the *aopN* coding sequences were cloned and fused with pBI121-3×FLAG (pBI121 inserted into 3×FLAG tag coding sequences, and the constructed primers are displayed in Appendix A), to produce the pBI121-3×FLAG-AopN. Mutant protein of AopN-LXXLL (amino acids R-366 and G-367 were mutated to AA) based on pBI121-3×FLAG-AopN vector was constructed using KOD-Plus-mutagenesis Kit (TOYOBO, Osaka, Japan). Mutant protein of AopN-DUF4129 (amino acids 20–66 coding sequences were deleted) and AopN-Cpta_toxin (amino acids 215-257 coding sequences were deleted) based on pBI121-3×FLAG-AopN vector was also constructed using KOD-Plus-mutagenesis Kit (TOYOBO, Osaka, Japan). All primers used in the PCR are listed in Appendix A.

### 4.4. Effector Confirmation of A. Citrulli AopN 

The full-length cDNA sequence and the promoter sequence of *aopN* were cloned and fused with the pBBRNolac-4×FLAG vector [17]. The construct was then conjugated into the *A. citrulli* WT strain Aac5 and its type III secretion system-defective *hrcJ* mutant by triparental mating. Effector secretion through the T3SS was confirmed as previously described [28]. WT strain Aac5 and its T3SS-deficient *hrcJ* mutant expressing the AopN-FLAG fusion protein were cultured in KB medium until the logarithmic growth phase and then cultured in the T3SS-inducing medium for 4 h. Subsequently, intracellular components and exocrine protein components were extracted. The proteins were precipitated from the T3SS-inducing medium and cell pellets with 10% trichloroacetic acid, and then separated by sodium dodecyl sulfate-polyacrylamide gel electrophoresis. The proteins were detected using the anti-Flag antibody (MBL, Beijing, China) by Western blotting. Anti-GAPDH was used as the internal reference protein. The experiment was independently repeated three times. All primers used in the PCR are listed in Appendix A.

### 4.5. Plant Infection Assays

For *A. citrulli* infection assays in watermelon, cotyledons (n = 6) of 2-week-old watermelon seedlings (cultivar Ruixin) were inoculated with WT Aac5 strain and *aopN* mutant at 1 × 10^4^ CFU/mL using a syringe. The *aopN*-complemented strain was used as the control. The cotyledons were photographed at 4 days after inoculation using an embedded operation system (EOS) 70D camera (Canon, Beijing, China). Quantitative analysis was then performed with inoculated watermelon cotyledons collected at 2 and 4 days, and bacterial populations were evaluated as described previously [42]. Each experiment was independently repeated three times.

For the *A. citrulli* infection assay in *N. benthamiana*, a qualitative analysis was performed using six plants with 12 different leaves of 4–6-week-old *N. benthamiana,* which were inoculated with WT Aac5, *aopN* mutant, and its complemented strain at 1 × 10^6^ CFU/mL using a syringe. The *hrcJ* mutant was used as the control. The leaves were photographed at 3 days after inoculation using an EOS 70D camera (Canon, Beijing, China) to record symptoms. Each experiment was independently repeated three times.

### 4.6. Flg22-Inudced ROS Burst Assay

The pBI121-3×FLAG vector inserted into 3 × FLAG tag to fuse with the pBI121 vector [43] was used for this assay. The full-length cDNA sequence of *AopN* was cloned into the pBI121-eGFP vector. The Flg22-elicited ROS assay was performed with the transient expression of EV (empty vector) or AopP fused FLAG using *A. tumefaciens* GV3101 at an optical density at 600 nm (OD_600_) of 0.5. The 4–6-week-old *N. benthamiana* leaves were used in this assay. Ten leaf disks (4 mm in diameter) were collected from each inoculated site 24 h after inoculation and floated in 100 μL of sterile distilled water in a 96-well plate. Next, the water was replaced with 100 μL of solution containing 100 nM flg22, 20 μg/mL horseradish peroxidase, and 100 μM luminol. Luminescence was recorded immediately using a Tecan Infinite F200 luminometer (Tecan, Mannedorf, Switzerland) for 40 min. The experiment was independently repeated three times. All primers used in the PCR are listed in Appendix A.

### 4.7. BiFC Assay

To generate BiFC constructs, the full-length cDNA of *AopN* was cloned into a pSPYNE^®^173 vector [44], and the full-length cDNA of *ClHIPP* and *ClLTP* was cloned into the pSPYCE (M) vector [44]. The constructs were co-expressed in *N. benthamiana* leaves for analysis, as previously described [44,45]. After 48 h, at an excitation wavelength of 488 nm to acquire fluorescent images, the plasma membrane marker dye (Dil, D3911, Thermo Scientific, Waltham, MA, USA) was used as the control for PM localization. For Dil staining, the *N. benthamiana* leaves were injected with 1 μM of Dil solution for 20 min. Then, the fluorescence signal was visualized under a Zeiss LSM 880 laser confocal microscope (Zeiss, Jena, Germany). Yellow fluorescent protein imaging was performed at an excitation wavelength of 513 nm. The Dil imaging was performed at an excitation wavelength of 550 nm. The experiment was independently repeated three times. All primers used in the PCR are listed in Appendix A.

### 4.8. Subcellular Localization of AopN 

The pBI121-eGFP vector inserted into eGFP tag to fuse with the pBI121 vector [43] was used for this assay. AopN was fused with the pBI121-eGFP vector for the subcellular localization assay, transformed into *A. tumefaciens* GV3101 strain, and the construct was transiently expressed in *N. benthamiana* at an OD_600_ of 0.3 by injection. After 36 h, the inoculated leaves were visualized under a Zeiss LSM 880 laser confocal microscope (Zeiss, Jena, Germany) at an excitation wavelength of 488 for GFP imaging or 588 nm for RFP imaging [46]. The plasma membrane marker (name as PM-rk), carrying red fluorescent protein tag was used as the control for PM localization [47]. The leaves injected with *A*. *tumefaciens* GV3101 strain carrying 35S: GFP were used as the control. The experiment was independently repeated three times.

### 4.9. Electrolyte Leakage Quantification

Electrolyte leakage quantification in *N. benthamiana* was based on the method of Stock et al. [8], slightly modified. For *N. benthamiana,* transient expression of EV (carrying the pBI121-3FLAG empty vector) or AopN-FLAG was performed in different regions of the same leaf using *A. tumefaciens* GV3101 at OD_600_ = 0.5. Four leaf disks (12 mm in diameter) were sampled from each inoculation region at 24 h post inoculation, rinsed in distilled water, and floated on 5 mL of distilled water in a six-well tissue plate under continuous light. At 24 h after initial *A. tumefaciens* inoculation, conductivity of the bathing water was measured using a conductivity meter (Leici, Shanghai, China) and repeated every 24 h. Mean conductivity ± standard deviation (SD) from two different leaves of three plants are reported for each treatment. The experiment was independently repeated three times.

### 4.10. PCD Phenotype in N. Benthamiana After Transient Expression of Empty Vector (EV) or AopN

For *N. benthamiana,* transient expression of EV (empty vector; carrying pBI121-3FLAG) or AopN-FLAG was performed in different regions of the same leaf using *A. tumefaciens* GV3101 at OD_600_ = 0.5. Ten different *N. benthamiana* plants were injected in the same way. Inoculated leaf tissue was monitored for 3 days, and measured after 3 days.

### 4.11. Expression Analysis of mRNA

The cotyledons of 2-week-old watermelon seedlings were sprayed with 0.1 μM flg22*^Ac^* (to indicate that this 22 peptide short amino acid sequence was from *A. citrulli*), 0.1 mM SA, and *A. citrulli* strain Aac5 at 3 × 10^8^ CFU/mL, and the total RNA was isolated from seedlings at different time points using the Quick-RNA Plant Kit (Zymo; cat. no. R2024), according to the manufacturer’s instructions. Subsequently qPCR was performed using the SYBR Green Real-Time PCR Master Mix (Toyobo, Japan). Gene expression levels of watermelon *ClHIPP* and *ClLTP* genes were measured. The watermelon *ACTIN* gene was used as the internal control [48]. EV and AopN-FLAG were transiently expressed by inoculating *A. tumefaciens* GV3101 strains into *N. benthamiana* leaves. After 24 h, the inoculated leaves were collected and floated in 5 mL of sterile distilled water in a six-well plate overnight. Next, water was replaced with 100 nM flg22 in each well. At the same time set, a group was processed without flg22 treament. The total RNA was extracted, and then the *NbPti5*, *NbAcre31*, and *NbGras2* gene expression levels were measured. The *N. benthamiana ACTIN* gene was used as the internal control [46]. All primers are shown in Appendix A. Each experiment was independently repeated three times.

### 4.12. Yeast Two-Hybrid Assay

To generate these constructs, the full-length cDNA of AopN was cloned into a pGBKT7 vector, and the full-length cDNA of *ClHIPP* and *ClLTP* was cloned into the pGADT7 vector. Subsequent co-transformation of the Gold2 strain, and set positive control (co-transformation of pGBKT7-53 and pGADT7-T into Gold2 strain) and negative control (co-transformation of pGBKT7-lam and pGADT7-T into Gold2 strain). The Gold2 yeast strain was transformed with the above constructs, and the yeast assay was performed as described previously [49]. Each experiment was independently repeated three times.

### 4.13. Statistical Analysis

Data were analyzed using independent-samples *t*-tests. Statistical analyses were conducted using SPSS version 17.0 (SPSS Inc., Chicago, IL, USA) and GraphPad PRISM 5.0 software (GraphPad Software Inc., La Jolla, CA, USA). Differences with a *p* value less than 0.05 were considered significant. 

## 5. Conclusions

In summary, AopN was identified as a type III secreted effector protein which induced PCD response. This protein was localized on the cell membrane and inhibited ROS bursts in plants. More importantly, the *A. citrulli* mutant strain lacking the *aopN* gene showed significantly reduced virulence in *N. benthamiana*. PCD-inducing ability was dependent on two key motifs. The effector protein AopN interacted with ClHIPP and ClLTP. The findings provide novel insights into how *A. citrulli* type III effector AopN interacts with its natural host watermelon as well as *N. benthamiana.*

## Figures and Tables

**Figure 1 ijms-21-06050-f001:**
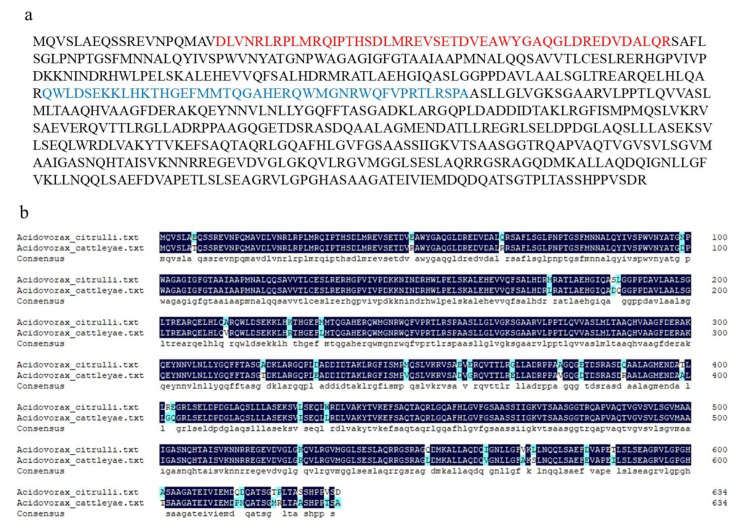
Sequence analysis of AopN from *Acidovorax citrulli*. (**a**) Amino acid sequence of AopN contained motifs DUF4129 (marked in red) and Cpta_toxin (marked in blue). (**b**) Pairwise sequence alignment of AopN with its homolog from *A. cattleyae* (WP_092832167).

**Figure 2 ijms-21-06050-f002:**
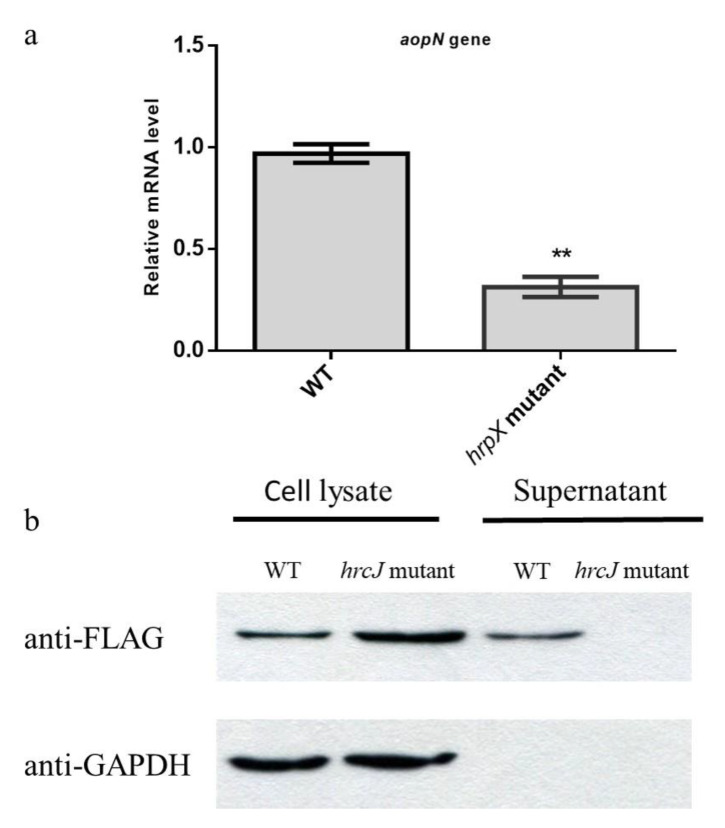
Analysis and identification of AopN as a type III secreted effector. (**a**) Quantitative PCR (qPCR) assays were performed to determine whether *aopN* was regulated by *hrpX.* The wild-type (WT) strain and its *hrpX* mutant were cultured in the T3SS-inducing medium. RNA was extracted and reverse transcribed into cDNA, which was subjected to qPCR assays. The *rpoB* was used as the internal reference gene [17]. The experiment was performed three times, with similar results obtained each time. Data represent the means ± SD (*n* = 3). ** above bars indicate significant differences as determined by t-test, *p* < 0.05; (**b**) Identification of AopN as a type 3 effector by Western blotting. WT strain Aac5 and its T3SS-deficient *hrcJ* mutant expressing the AopN-fused FLAG protein was cultured in KB medium until the logarithmic growth phase and then cultured in the T3SS-inducing medium for 4 h. Subsequently, intracellular components and exocrine protein components were extracted and detected using the anti-FLAG antibody. Signals were detected in the intracellular component of the WT and *hrcJ*, as well as the supernatant of the WT, but no signal was detected in the exocrine component of *hrcJ* mutant carrying the AopP-FLAG. The antibody GAPDH was used as the internal reference protein [17]. The experiment was performed three times, with similar results obtained each time.

**Figure 3 ijms-21-06050-f003:**
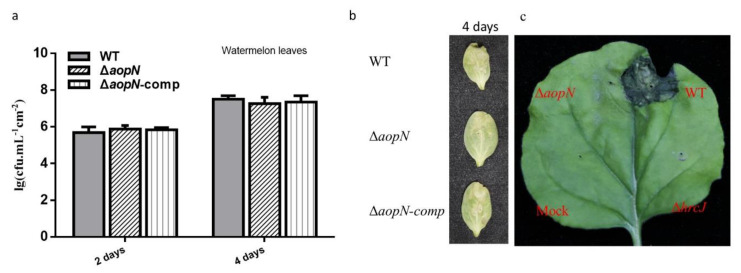
Virulence analysis of AopN. (**a**) Bacteria population levels of wild-type (WT) strain, *aopN* mutant, and its complemented strain in watermelon leaves. The strains were cultured until the logarithmic phase, resuspended in 10 mM MgCl_2_ at 1 × 10^4^ colony-forming units (CFU) /mL and injected into the watermelon leaves. Samples were collected at two and four days after injection and bacterial population levels were measured. (**b**) Qualitative virulence analysis of AopN in watermelon. The strains were cultured until the logarithmic phase, resuspended in 10 mM MgCl_2_ at 1 × 10^4^ CFU/mL, and injected into the watermelon leaves. Photographs were captured four days after inoculation to record the symptoms. (**c**) Qualitative virulence analysis of AopN in *N. benthamiana*. The strains were cultured until the logarithmic phase, resuspended in 10 mM MgCl_2_ at 1 × 10^6^ CFU/mL and injected into *N. benthamiana* leaves. Photographs were captured three days after inoculation to record the symptoms. Three independent assays were conducted in each instance, with similar results obtained each time. At least six plants with 12 different leaves were observed.

**Figure 4 ijms-21-06050-f004:**
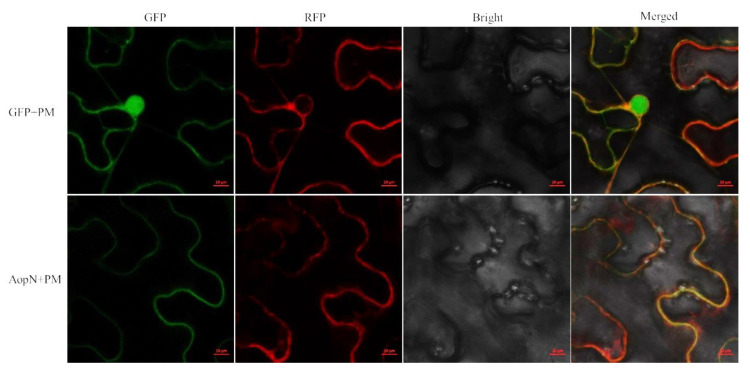
Subcellular localization of AopN. The leaves injected with *Agrobacterium tumefaciens* GV3101 strain carrying 35S: AopN-fused green fluorescent protein (GFP) were sampled and observed under a confocal microscope (20 ×). A plasma membrane (PM) marker carrying red fluorescent protein (RFP) tag was used as the control for PM localization. The leaves injected with *A*. *tumefaciens* GV3101 strain carrying 35S: GFP were used as the control. The experiment was performed three times, with similar results obtained each time. Scale bar, 10 µm.

**Figure 5 ijms-21-06050-f005:**
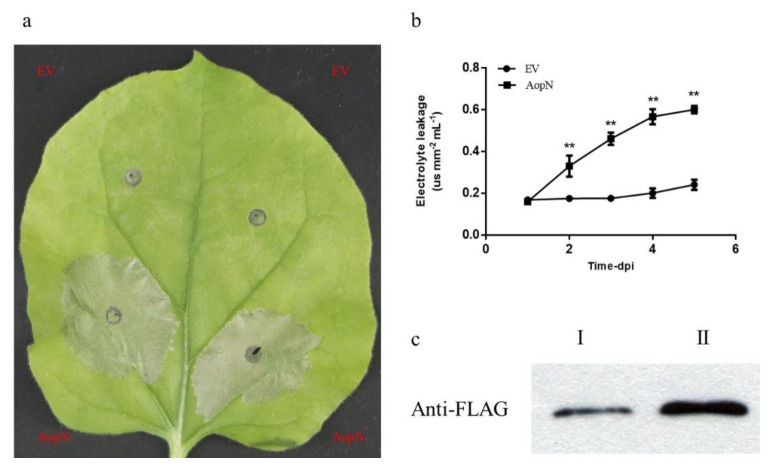
AopN induced program cell death (PCD) in *N. benthamiana.* (**a**) AopN induced PCD symptoms in *N. benthamiana.* The *N. benthamiana* leaves injected with *Agrobacterium tumefaciens* GV3101 strain carrying 35S: AopN-FLAG or the empty vector (EV) control were sampled 3 days after inoculation. The experiment was performed three times, with similar results obtained each time. Three independent assays were conducted and at least 10 plants with 20 different leaves were observed. (**b**) Conductivity test with AopN. The *N. benthamiana* leaves were injected with *A. tumefaciens* GV3101 strain carrying 35S: AopN-FLAG or the empty vector. Leaf disks at the injection site were removed with a puncher and placed in 5 mL sterile water in a six-well dish. After resting for 24 h, conductivity levels were measured using a conductivity meter (DDS-307, Leici Experimental Instrument Factory, Shanghai China). The experiment was performed three times, with similar results obtained each time. Data represent the means ± SD (*n* = 6). ** above bars indicate significant differences as determined by t-test, *p* < 0.05. (**c**) Analysis of AopN protein secretion. The leaves at the injection site were collected and proteins were detected by Western blotting. I represents the 36 h sample, II represents the 48 h sample. The experiment was performed three times, with similar results obtained each time.

**Figure 6 ijms-21-06050-f006:**
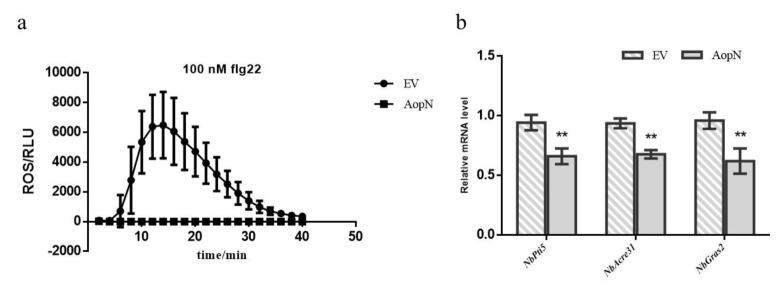
AopN inhibited pathogen-associated molecular pattern-triggered immunity (PTI) responses. (**a**) AopN inhibited pathogen-associated molecular pattern (PAMP) flg22-induced reactive oxygen species (ROS) burst in *N. benthamiana. Agrobacterium tumefaciens* strain GV3101 carrying AopN-FLAG reduced ROS production induced by flg22 in *N. benthamiana* compared with that carrying the EV (empty vector) controls. The experiment was performed three times, with similar results obtained each time. Data represent the means ± SD (*n* = 10). **(b)** Expression levels of the PTI markers: *NbPti5*, *NbAcre31*, and *NbGras2* with flg22 treatment. The experiment was performed three times, with similar results obtained each time. Data represent the means ± SD (*n* = 3). ** above bars indicate significant differences as determined by t-test, *p* < 0.05

**Figure 7 ijms-21-06050-f007:**
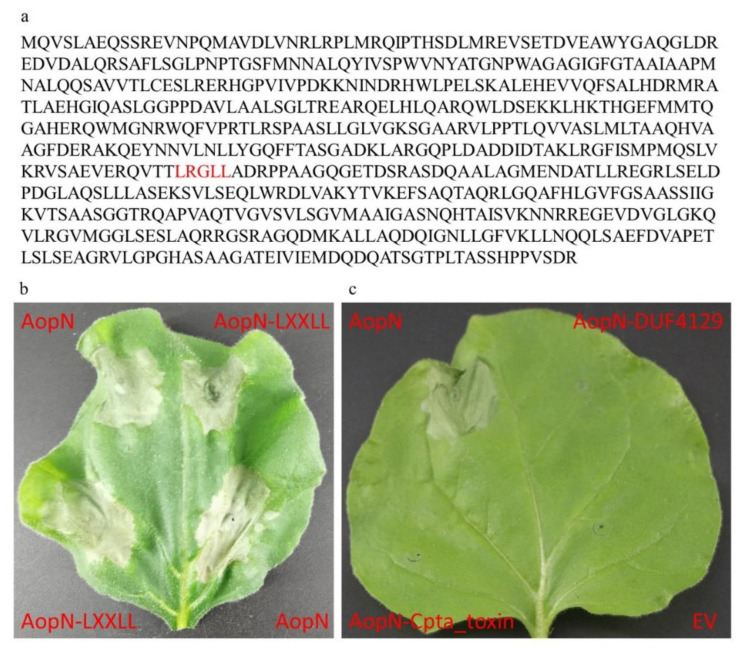
Analysis of the key motifs of AopN-induced programmed cell death (PCD). **(a)** Red font is the LXXLL motif prediction in AopN. (**b**) PCD symptoms induced by AopN-LXXLL. (**c**) Lack of PCD symptoms induced by AopN lacking DUF4129 and Cpta_toxin motifs. The experiments were performed three times, with similar results obtained each time. *A. tumefaciens* GV3101 carrying EV as control. Three independent assays were conducted and at least five plants with 10 different leaves were observed.

**Figure 8 ijms-21-06050-f008:**
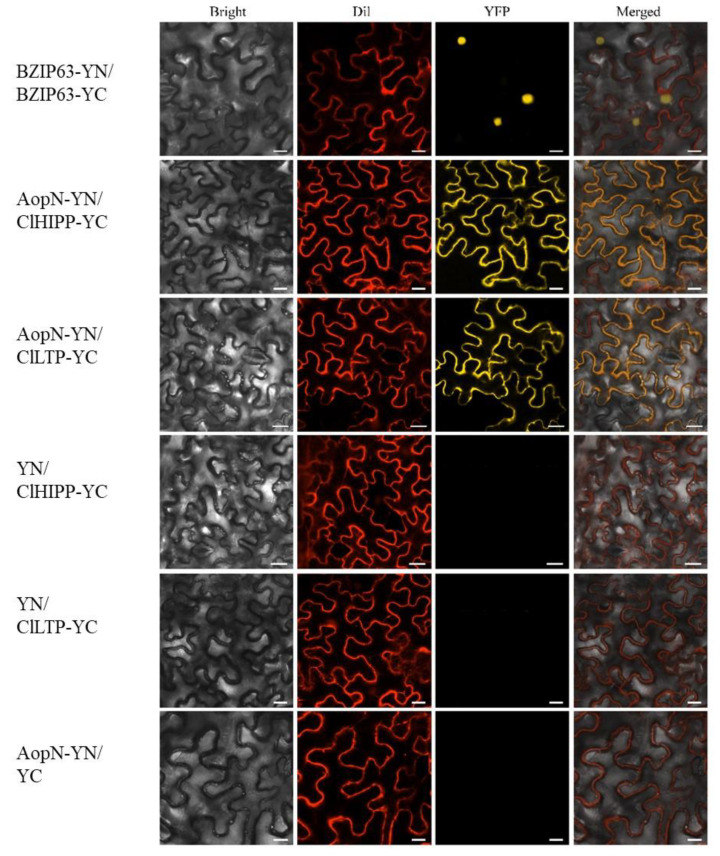
**The** bimolecular fluorescence complementation (BiFC) verified the interaction between AopN and ClHIPP, as well as AopN and ClLTP. AopN was fused with the nYFP tag, and ClHIPP and ClLTP were fused with the cYFP tag. Fluorescent dye (D3911, Thermo) was used as a control for plasma membrane localization. The experiment was performed three times, with similar results obtained each time. The white short line in the lower right corner of the figure represents the scale bar (25 µm).

**Figure 9 ijms-21-06050-f009:**
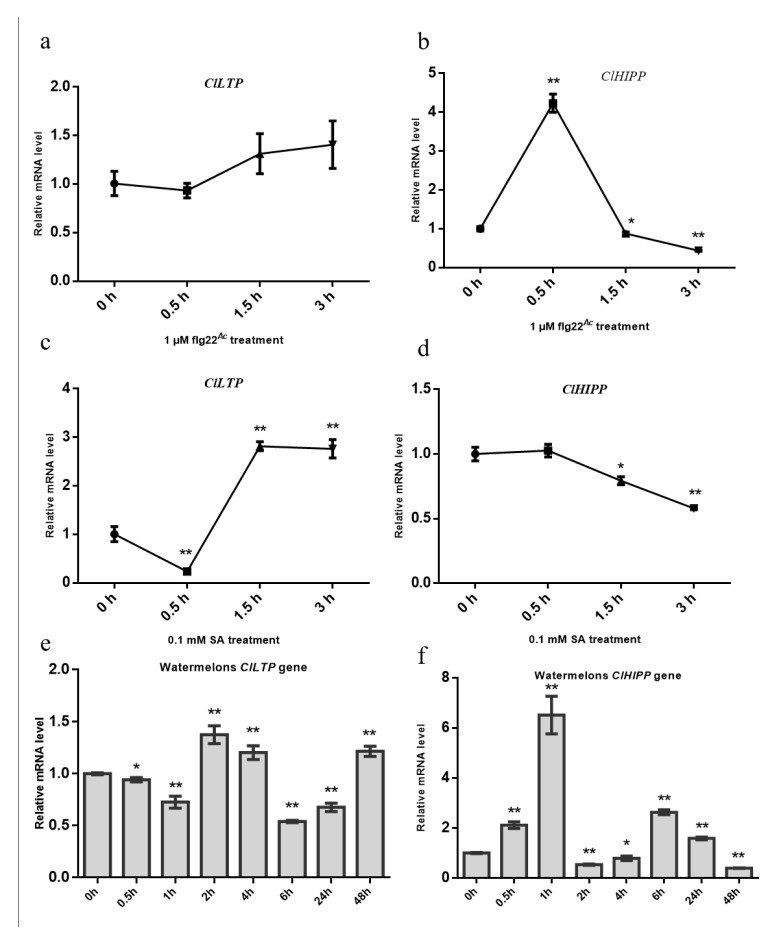
*ClHIPP* and *ClLTP* are involved in plant immunity. (**a**,**b**) Expression patterns of *ClHIPP* and *ClLTP* in watermelon at the indicated time points after flg22*^Ac^* treatment. (**c**,**d**) Expression patterns of *ClHIPP* and *ClLTP* in watermelon at the indicated time points after salicylic acid (SA) treatment. Watermelon *ACTIN* was used as the internal reference gene. The experiment was performed three times, with similar results obtained each time. (**e**,**f**) Expression patterns of *ClHIPP* and *ClLTP* in watermelon at the indicated time points after *A. citrulli* strain Aac5 infected. Data represent the means ± SD (*n* = 3). *, ** above bars indicate significant differences as determined by t-test, *p* < 0.05.

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
