# Peer review of "Identification and Functional Analysis of AopN, an Acidovorax Citrulli Effector that Induces Programmed Cell Death in Plants"

_ijms, 2020, doi:10.3390/ijms21176050_

Round 1

Reviewer 1 Report

Zhang et al. identified an effector AopN from A. citrulli and characterized its virulence function and localization. The authors also identified two interacting proteins, ClHIPP and ClLTP, by yeast two-hybrid screen using AopN as bait and confirmed the expression of the two interactors are altered by flg22 and SA. My specific comments are listed as follows:

  1. ROS can also be induced during early stage of PCD. Please describe how to discriminate the ROS induced by flg22 and effector A.
  2. The authors stated that both ClHIPP and ClLTP genes were induced by SA. But in Figure 9, the ClHIPP gene is downregulated, rather than induced by SA. It is suggested the authors include appropriate controls such as water application to confirm the conclusion.
  3. The involvement of ClHIPP and ClLTP in plant immunity is only based on the gene expression data. It would be convincing that if these two genes are silenced by VIGS followed by inoculation with Agraobactium carrying AopN and see whether AopN-induced cell death is affected.
  4. The authors need to pay attention to the use of Latin names and their abbreviations. Generally, the full name should be mentioned when it first appears in the text while only the abbreviations should be used thereafter, such as “Nicotiana benthamiana”, “pattern recognition receptor (PRR)”.
  5. Several grammar mistakes need to be corrected.
  • For example, in the Results session, line 230-232, ” As shown in Figure 7c, GV3101 strains without AopN-DUF4129 and AopN-Cpta_toxin motifs lost its ability to induce PCD….”. The description is not correct.
  • Line 115, “…hrcJ mutant expressing AopN-fused…”. It is suggested to use the format “AopN-FLAG” to indicate fusion proteins and the format “35S:GFP” to indicate promoters and genes.
  • Line 226, “As shown in Figure 7b, the strastrain expressing”.

Author Response

Comments and Suggestions for Authors

Zhang et al. identified an effector AopN from A. citrulli and characterized its virulence function and localization. The authors also identified two interacting proteins, ClHIPP and ClLTP, by yeast two-hybrid screen using AopN as bait and confirmed the expression of the two interactors are altered by flg22 and SA. My specific comments are listed as follows:

  1. ROS can also be induced during early stage of PCD. Please describe how to discriminate the ROS induced by flg22 and effector A.

Dear reviewer, thanks for your question. This question is very meaningful. Since the early stage of PCD in plant can trigger ROS (Vacca et al., 2004), for example ROS was triggered in rice during salt stress-induced early stage of PCD (Chen et al., 2009), it is necessary to find a suitable expression time point to detect this effect. The electrolyte leakage is an important indicator to measure the PCD response of plants (Kacprzyk et al., 2016). In a previous study, the electrolyte leakage assay was used for detecting the PCD induced by XopX in N.benthamiana (Stork et al., 2015). In current study, AopN induced the electrolyte leakage of N. benthamiana at 24 h showed no significant difference compared EV control with using A. tumefaciens GV3101 at an optical density at 600 nm (OD600) of 0.5 (Figure 5b). And at this point in time, the AopN fused FLAG can be expressed in N.benthamiana by western blot (Figure 5c). Therefore, it is reasonable to test whether AopN affects ROS under this condition (24 h, A. tumefaciens GV3101 at an optical density at OD600 of 0.5.). We also made corresponding modification in the manuscript.

2. The authors stated that both ClHIPP and ClLTP genes were induced by SA. But in Figure 9, the ClHIPP gene is downregulated, rather than induced by SA. It is suggested the authors include appropriate controls such as water application to confirm the conclusion.

Dear reviewer, thanks for your question. The plant hormones such as salicylic acid (SA), are considered to play critical roles in various aspects of the plant immune system (Pieterse et al., 2009;Pieterse et al., 2012). Especially for some plant pathogenic bacteria, the effector always interferes with SA signaling to suppress plant immunity, such as effector XopJ (Ustun et al., 2013), effector AvrPto1B (Chen et al., 2017). As the immune response of watermelon is rarely reported, a recent paper in watermelon mentioned that SA is related to resistance to nematode infestation (Yang et al., 2018). Therefore, in this study we used SA to treat watermelon to determine whether the two molecular targets (ClLTP, ClHIPP) are associated with the SA pathway. I’m sorry that I did not clarify my meaning clearly here. We should use “associate with SA” instead of “induce SA”. In fact, we are not emphasizing SA, but want to show that these two targets whether the expression of ClHIPP and ClLTP responds to A.citrulli strain Aac5 infection. In order to prove that these two targets responds to A.citrulli strain Aac5 infection at the transcription level, we added an experiment. We used the wild-type strain Aac5 to stimulate the watermelon. In the range of 8 time points, we detected the transcriptional expression of the two targets, combined with the induction experiment of SA and flg22, Comprehensively judge whether the two targets responds to A.citrulli strain Aac5 infection at the transcription level. The method of this experiment design is based on the report on Cheng et al.(2016). In this report (Cheng et al., 2016), SA, flg22 and wild-type pathogens are also used to treat cotton. Their controls are all 0 h as the starting point, so we also adopted the 0 h control treatment. Based on this successful case, we can determine whether the two targets responds to A.citrulli strain Aac5 infection at the transcription level. Thank you very much for your suggestions, we will consider your suggestions in future experiments. We also made corresponding modification in the manuscript.

3. The involvement of ClHIPP and ClLTP in plant immunity is only based on the gene expression data. It would be convincing that if these two genes are silenced by VIGS followed by inoculation with Agraobactium carrying AopN and see whether AopN-induced cell death is affected.

Dear reviewer, thanks for your question. This is a good idea. This suggestion is very helpful for our future work. There have been few reports of effector in A.citrulli, which is mainly limited by lack of a suitable plant interaction system. We have previously shown that hrpG and hrpX are the core regulators of T3SS (Zhang et al., 2018), providing a foundation for the screening and identification of T3Es. Recently, a screening system for T3SS inhibitors in A. citrulli was established (Ma et al., 2019), which will also be useful for screening of T3E. In addition, some potential candidate T3Es were also screened based on differences in transcription between A. citrulli wild-type (WT) strain M6 and its hrpX mutant as well as using transcriptome analysis (Jimenez-Guerrero et al., 2020). A recent study has confirmed that the model plant Nicotiana benthamiana can be used to analyze the pathogenic mechanism of A.citrulli (Traore et al., 2019). In the past 5-6 years, studies of T3Es in A.citrulli have been progressing, from the initial analysis of homologous effector proteins (Eckshtain-Levi et al., 2014), and the possible relationship between effector proteins and host preference (Yan et al., 2017), to the novel T3Es identifications (Jimenez-Guerrero et al., 2020). However, few studies have been performed to identify and determine the function of A.citrulli effectors. In fact, there is currently no report to reveal the mechanism of the interaction between the effector in A.citrulli and the natural host watermelon. The application of VIGS technology in watermelon is still immature and there are few reports. In the past, we have made some attempts using transient expression in watermelon, but the effect is very bad, and the transient expression of watermelon protoplasts is currently immature and the expression is relatively poor. Therefore, it is difficult to realize this perfect idea with the current technical conditions. In addition, in the answer to question two, we also added the experiment of A.citrulli treating watermelon. Using SA, flg22 and pathogen treatment experiments to supplement our conclusions. However, in future research, we will also work hard to break through technical barriers. In future work, I believe there will be more reports of watermelon that are worth learning. This provides the possibility for us to analyze the function of the target in depth. Thank you very much for your suggestion, this is a good good good idea.

4. The authors need to pay attention to the use of Latin names and their abbreviations. Generally, the full name should be mentioned when it first appears in the text while only the abbreviations should be used thereafter, such as “Nicotiana benthamiana”, “pattern recognition receptor (PRR)”.

Dear reviewer, thanks for your question. The word “Nicotiana benthamiana” appears in line 18 for the first time, and the following is abbreviated “N.benthamiana” according to your suggestions. This sentence “by the pattern recognition receptor (PRR) on the plant cell surface” has been revised in line 34-35 according to your suggestions. In addition, the same error we also found exists in line 36-37, “PR gene” has been modified to “pathogenesis-related (PR) gene”. We will pay attention to this detail in future writing, thank you for your suggestions.

5. Several grammar mistakes need to be corrected.

  • For example, in the Results session, line 230-232, ” As shown in Figure 7c, GV3101 strains without AopN-DUF4129 and AopN-Cpta_toxin motifs lost its ability to induce PCD….”. The description is not correct.
  • Line 115, “…hrcJ mutant expressing AopN-fused…”. It is suggested to use the format “AopN-FLAG” to indicate fusion proteins and the format “35S:GFP” to indicate promoters and genes.
  • Line 226, “As shown in Figure 7b, the strastrain expressing”.

Dear reviewer, thanks for your question. There are some problems in some details in the manuscript. In order to better solve this problem, we have modified the language problem in the manuscript and also made these according to your requirements. Several grammar mistakes has been revised in this manuscript, if you have any questions, please feedback to us in time, thank you very much for your suggestions.

References

Chen, H., Chen, J., Li, M., Chang, M., Xu, K., Shang, Z., Zhao, Y., Palmer, I., Zhang, Y., Mcgill, J., Alfano, J.R., Nishimura, M.T., Liu, F., and Fu, Z.Q. (2017). A Bacterial Type III Effector Targets the Master Regulator of Salicylic Acid Signaling, NPR1, to Subvert Plant Immunity. Cell Host & Microbe.

Chen, X., Wang, Y., Li, J., Jiang, A., Cheng, Y., and Zhang, W. (2009). Mitochondrial Proteome During Salt Stress-Induced Programmed Cell Death in Rice. Plant Physiol Biochem 47, 407-415.

Cheng, H.Q., Han, L.B., Yang, C.L., Wu, X.M., Zhong, N.Q., Wu, J.H., Wang, F.X., Wang, H.Y., and Xia, G.X. (2016). The Cotton MYB108 Forms a Positive Feedback Regulation Loop with CML11 and Participates in the Defense Response against Verticillium dahliae Infection. J Exp Bot 67, 1935-1950.

Eckshtain-Levi, N., Munitz, T., Zivanovic, M., Traore, S.M., Sproer, C., Zhao, B., Welbaum, G., Walcott, R., Sikorski, J., and Burdman, S. (2014). Comparative Analysis of Type III Secreted Effector Genes Reflects Divergence of Acidovorax citrulli Strains into Three Distinct Lineages. Phytopathology 104, 1152-1162.

Jimenez-Guerrero, I., Perez-Montano, F., Da Silva, G.M., Wagner, N., Shkedy, D., Zhao, M., Pizarro, L., Bar, M., Walcott, R., Sessa, G., Pupko, T., and Burdman, S. (2020). Show Me Your Secret(Ed) Weapons: A Multifaceted Approach Reveals a Wide Arsenal of Type III-Secreted Effectors in the Cucurbit Pathogenic Bacterium Acidovorax citrulli and Novel Effectors in the Acidovorax Genus. Mol Plant Pathol 21, 17-37.

Kacprzyk, J., Dauphinee, A.N., Gallois, P., Gunawardena, A.H., and Mccabe, P.F. (2016). Methods to Study Plant Programmed Cell Death. Methods Mol Biol 1419, 145-160.

Ma, Y.N., Chen, L., Si, N.G., Jiang, W.J., Zhou, Z.G., Liu, J.L., and Zhang, L.Q. (2019). Identification of Benzyloxy Carbonimidoyl Dicyanide Derivatives as Novel Type III Secretion System Inhibitors Via High-Throughput Screening. Front Plant Sci 10, 1059.

Pieterse, C.M., Leon-Reyes, A., Van Der Ent, S., and Van Wees, S.C. (2009). Networking by Small-Molecule Hormones in Plant Immunity. Nat Chem Biol 5, 308-316.

Pieterse, C.M., Van Der Does, D., Zamioudis, C., Leon-Reyes, A., and Van Wees, S.C. (2012). Hormonal Modulation of Plant Immunity. Annu Rev Cell Dev Biol 28, 489-521.

Stork, W., Kim, J.G., and Mudgett, M.B. (2015). Functional Analysis of Plant Defense Suppression and Activation by the Xanthomonas Core Type III Effector XopX. Molecular plant-microbe interactions : MPMI 28, 180.

Traore, S.M., Eckshtain-Levi, N., Miao, J., Castro Sparks, A., Wang, Z., Wang, K., Li, Q., Burdman, S., Walcott, R., Welbaum, G.E., and Zhao, B. (2019). Nicotiana Species as Surrogate Host for Studying the Pathogenicity of Acidovorax citrulli, the Causal Agent of Bacterial Fruit Blotch of Cucurbits. Mol Plant Pathol.

Ustun, S., Bartetzko, V., and Bornke, F. (2013). The Xanthomonas Campestris Type III Effector XopJ Targets the Host Cell Proteasome to Suppress Salicylic-Acid Mediated Plant Defence. PLoS Pathog 9, e1003427.

Vacca, R.A., De Pinto, M.C., Valenti, D., Passarella, S., Marra, E., and De Gara, L. (2004). Production of Reactive Oxygen Species, Alteration of Cytosolic Ascorbate Peroxidase, and Impairment of Mitochondrial Metabolism Are Early Events in Heat Shock-Induced Programmed Cell Death in Tobacco Bright-Yellow 2 Cells. Plant Physiol 134, 1100-1112.

Yan, L., Hu, B., Chen, G., Zhao, M., and Walcott, R.R. (2017). Further Evidence of Cucurbit Host Specificity among Acidovorax citrulli Groups Based on a Detached Melon Fruit Pathogenicity Assay. Phytopathology 107, 1305-1311.

Yang, Y.-X., Wu, C., Ahammed, G.J., Wu, C., Yang, Z., Wan, C., and Chen, J. (2018). Red Light-Induced Systemic Resistance against Root-Knot Nematode Is Mediated by a Coordinated Regulation of Salicylic Acid, Jasmonic Acid and Redox Signaling in Watermelon. Frontiers in Plant Science 9, 899.

Zhang, X., Zhao, M., Yan, J., Yang, L., Yang, Y., Guan, W., Walcott, R., and Zhao, T. (2018). Involvement of hrpX and hrpG in the Virulence of Acidovorax citrulli Strain Aac5, Causal Agent of Bacterial Fruit Blotch in Cucurbits. Frontiers in Microbiology 9.

Reviewer 2 Report

The manuscript by Xiaoxiao Zhang and colleagues aims at demonstrating that AopN is a type 3 effector in A. citrulli strain Aac5 and contributes to virulence in N. benthamiana but not in watermelon. In the attempt to characterize the function of AopN, authors show that the protein is localized to plasma membrane (PM), induces cell death (PCD) response in N. benthamiana leaves and interacts with ClHIPP and ClLTP in the natural host plant, watermelon.

Overall, I find that the paper describes well the results of the experiments which are appropriate to reach the conclusions claimed by authors. Nevertheless, I have a few major concerns that I think the authors should address:

  1. AopN interacted with ClHIPP and ClLTP: the bimolecular fluorescence complementation (BiFC) assay showed in figure 8 looks convincing to me, even if in the AopN-YN/ClHIPP-YC combination cytosolic strands are visible, where both AopN-YN/ClHIPP-YC and the fluorescent dye localized. All three of them are supposed to be PM-localized. Can the authors explain or try to speculate on this? Authors should (must?) also provide more information on the nature of the fluorescent dye used in those experiments. I think that the conclusion on the in vivo direct interaction between the AopN and its targets should be supported by adopting a second and different method, such as coIP-based pull-down assays or copurification as described in Jung-Gun Kim et al., 2009. In line 243 authors claim that the above cited targets of AopN were identified by yeast two-hybrid screening. Authors should add those results to the manuscript. Since DUF4129 and Cpta_toxin motifs are shown to play important roles in regulating PCD in N. benthamiana, it would be interesting to know whether AopN versions containing mutations in those motifs maintain the ability to interact with the effector targets.  
  1. AopN suppressed ROS burst in benthamiana: results shown in figure 6 (a) focused on ROS burst analyses following flg22 treatment should be correlated to the expression of the effector. Maybe (q)PCR analyses are appropriate and sufficient. Analyses on gene expression showed in figure 6 (b) should be done in the presence or absence of flg22, to asses that this PTI related response is inhibited by AopN.
  2. More details on the Zeiss 880 confocal setup and the exact settings of the wavelength windows used to collect the signals as well as on the PM marker carrying red fluorescent protein tag used as the control for PM localization must be provided.

 Minor concerns:

  1. In line 328 authors claim that PTI and ETI are independent of each other. This doesn’t consider the conclusions of two papers recently published by Minhang Yuan et al., 2020 and Bruno Pok Man Ngou et al., 2020. Authors should include these papers in the discussion part.
  2. I found problems with the space between words throughout the entire manuscript.

Reviewer 3 Report

The work of Zhang and colleagues investigates the mechanism of infection of Acidovorax citrulli.

They identify the effector AopNas as an important factor for pathogenesis and proceed to study its mechanism using Nicotiana bentamiana. They found that AopN can inhibit reactive oxygen burst in the flg22-assay and can trigger program cell death.

They then delineate a possible pathway for its function, which comprises the interaction with two novel factors, ClHIPP and ClLTP.

Overall the work is logically presented and the experiments proposed represent a rigorous approach for the understanding of the mechanism behind the virulence of pathogen. The use ofNicotianais well justified by the difficulty of working with the species targeted by the pathogen, like watermelon.

However it is of particular importance that the results show the contribution in virulence of AopN in Nicotiana, PCD and ROS burst, but not in watermelon.

Besides minor text errors I suggest that in the discussion and conclusion the author clearly delineate the reason for this difference in behaviour of the pathogen in Nicotianarespect to watermelon, in particular considering that the AopN interactors, ClHIPPand ClLTPare induced by flg22AC.

In conclusion I consider the paper presents some interesting findings, which contributes to the understanding of the biological pathway of an agronomical important pathogen.

Author Response

  1. The work of Zhang and colleagues investigates the mechanism of infection of Acidovorax citrulli.

Dear reviewer, thanks for your question. This question is a great encouragement to us. There are only a few studies on effector proteins, and their functions in Acidovorax citrulli, a plant pathogen that affects watermelon and other cucurbitaceous crops. In this study, we identified the effector protein AopN, which localizes to the cell membrane and induces programmed cell death in Nicotiana benthamiana. More importantly, the pathogenicity of N. benthamiana was significantly reduced when the effector protein coding gene was knocked out. This study provides insight into the pathogenic mechanism of A. citrulli. We believe that our study makes a significant contribution to the literature because these findings indicate that AopN suppresses plant immunity and triggers the effector-triggered immunity pathway. Thank you very much. In fact, the study of effector of A.citrulli is very difficult due to a severe lack of references. We will work harder in future work to reveal more pathogenic mechanisms of this bacteria.

  1. They identify the effector AopN as as an important factor for pathogenesis and proceed to study its mechanism using Nicotiana bentamiana. They found that AopN can inhibit reactive oxygen burst in the flg22-assay and can trigger program cell death.

Dear reviewer, thanks for your question. This question is a great encouragement to us. As the immune response of watermelon is rarely reported and references are lacking, we selected model plant N.benthamiana to study its pathogenic mechanism. In fact, a recent study has confirmed that the model plant Nicotiana benthamiana can be used to analyze the pathogenic mechanism of A.citrulli (Traore et al., 2019). ROS burst is a key event in the PTI response (Wei et al., 2018). For many effectors, whether it affects ROS is a very important research indicator, for example AvrRxo1(Shidore et al., 2017). In the current study, we found that AopN can inhibit the ROS outbreak, and can induce PCD, and more importantly, it is essential for the pathogenic ability of N.benthamiana. Thank you very much for your recognition and encouragement of our work.

  1. They then delineate a possible pathway for its function, which comprises the interaction with two novel factors, ClHIPP and ClLTP.

Dear reviewer, thanks for your question. This question is a great encouragement to us. In the case of using model plants to understand its suppression of plant immunity, we are very concerned about how it interferes with the immune response of the natural host watermelon. Therefore, in the current study, we have screened and found two molecular targets. This is very important for us to further reveal the pathogenesis of its natural host watermelon. Thank you very much for your recognition and encouragement of our work.

  1. Overall the work is logically presented and the experiments proposed represent a rigorous approach for the understanding of the mechanism behind the virulence of pathogen. The use of Nicotianais well justified by the difficulty of working with the species targeted by the pathogen, like watermelon.

Dear reviewer, thanks for your question. This question is a great encouragement to us. There are few reports on the immune response of watermelon, and it is not yet mature to use watermelon to study its pathogenic mechanism. In fact, in the past few years, we have been looking for suitable alternative hosts. At first, we tried Arabidopsis, but A.citrulli was not able to infect with Arabidopsis. We turned to tobacco. Due to transient expression technology and VIGS technology is very mature in N.benthamiana, so it will play an important role in the future research.

  1. However it is of particular importance that the results show the contribution in virulence of AopN in Nicotiana, PCD and ROS burst, but not in watermelon.

Dear reviewer, thanks for your question. This question is a great encouragement to us. In the current study, AopN can indeed inhibit ROS, induce PCD, and more importantly, it participates in the pathogenic ability of Ac on N.benthamiana, but not watermelon. This shows that Ac has other effectors similar to AopN in the process of infecting watermelon, and there is redundancy in function.

  1. Besides minor text errors I suggest that in the discussion and conclusion the author clearly delineate the reason for this difference in behaviour of the pathogen in Nicotiana respect to watermelon, in particular considering that the AopN interactors, ClHIPP and ClLTP are induced by flg22AC.

Dear reviewer, thanks for your question.

In the discussion, we discussed the difference between watermelon and N.benthamiana. It is well known that the functions of many T3Es in phytopathogenic bacteria are redundant. For instance, the T3Es AvrPto and AvrPtoB from Pseudomonas syringae pv. tomato DC3000 (Pst DC3000) strain have redundant functions [29]. Therefore, when a WT strain of phytopathogenic bacteria lacks a single effector, the virulence may not be substantially altered. For example, the deletion of only the avrBs2  affected the pathogenicity of X. oryzae pv. oryzicola strain RS105, whereas deleting other effector-coding genes did not affect its pathogenicity in rice [14]. In this study, AopN was found to be a key contributor to virulence in N. benthamiana (Figure 3c). In a previous study, the xopN mutant did not affect the virulence of X. oryzae pv. oryzicola strain RS105 in rice [14], but xopN was a key contributor to the virulence of X. oryzae pv. oryzicola strain GX01 [30]. In the current study, the aopN mutant did not affect the pathogenicity of A. citrulli strain Aac5 in its natural host watermelon (Figure 3a, b), suggesting that A. citrulli may have other functional redundant proteins during watermelon infection. In a previous study, the XopN homolog was found to be functionally redundant with XopZ and XopV [31]. These findings also indicate that the interaction between AopN and N. benthamiana is important to elucidate the pathogenic mechanisms of A. citrulli. If I don’t understand your question, please tell me in time so that I can modify it accordingly. Thank you for your suggestion.

  1. In conclusion I consider the paper presents some interesting findings, which contributes to the understanding of the biological pathway of an agronomical important pathogen.

Thank you for your approval. In fact, it is true. The current research on the effector of plant pathogenic bacteria focuses more on model bacteria, such as Pseudomonas and Xanthmonas. The attention to Ac is very low, but BFB triggered by Ac causes a lot of economic losses to watermelon. There are very few references in the research process of Ac and there are many technical obstacles. Therefore, more reports are needed to reveal the pathogenic mechanism of this pathogen.

References

Shidore, T., Broeckling, C.D., Kirkwood, J.S., Long, J.J., Miao, J., Zhao, B., Leach, J.E., and Triplett, L.R. (2017). The Effector AvrRxo1 Phosphorylates Nad in Planta. PLoS Pathog 13, e1006442.

Traore, S.M., Eckshtain-Levi, N., Miao, J., Castro Sparks, A., Wang, Z., Wang, K., Li, Q., Burdman, S., Walcott, R., Welbaum, G.E., and Zhao, B. (2019). Nicotiana Species as Surrogate Host for Studying the Pathogenicity of Acidovorax citrulli, the Causal Agent of Bacterial Fruit Blotch of Cucurbits. Mol Plant Pathol.

Wei, H.L., Zhang, W., and Collmer, A. (2018). Modular Study of the Type III Effector Repertoire in Pseudomonas syringae pv. tomato DC3000 Reveals a Matrix of Effector Interplay in Pathogenesis. Cell Rep 23, 1630-1638.

Round 2

Reviewer 2 Report

The authors have sufficiently addressed my concerns. Therefore, I'd recommend to accept the revised version of the manuscript for publication.